# The IDO Metabolic Trap Hypothesis for the Etiology of ME/CFS

**DOI:** 10.3390/diagnostics9030082

**Published:** 2019-07-26

**Authors:** Alex A. Kashi, Ronald W. Davis, Robert D. Phair

**Affiliations:** 1Stanford Genome Technology Center, Stanford University, Palo Alto, CA 94304, USA; 2Departments of Biochemistry and Genetics, Stanford University, Stanford, CA 94305, USA; 3Integrative Bioinformatics Inc., Mountain View, CA 94041, USA

**Keywords:** tryptophan metabolism, indoleamine-2,3-dioxygenase, bistability, kynurenine pathway, substrate inhibition, myalgic encephalomyelitis, chronic fatigue syndrome, mathematical model, critical point

## Abstract

Myalgic encephalomyelitis/chronic fatigue syndrome (ME/CFS) is a debilitating noncommunicable disease brandishing an enormous worldwide disease burden with some evidence of inherited genetic risk. Absence of measurable changes in patients’ standard blood work has necessitated ad hoc symptom-driven therapies and a dearth of mechanistic hypotheses regarding its etiology and possible cure. A new hypothesis, the indolamine-2,3-dioxygenase (IDO) metabolic trap, was developed and formulated as a mathematical model. The historical occurrence of ME/CFS outbreaks is a singular feature of the disease and implies that any predisposing genetic mutation must be common. A database search for common damaging mutations in human enzymes produces 208 hits, including IDO2 with four such mutations. Non-functional IDO2, combined with well-established substrate inhibition of IDO1 and kinetic asymmetry of the large neutral amino acid transporter, LAT1, yielded a mathematical model of tryptophan metabolism that displays both physiological and pathological steady-states. Escape from the pathological one requires an exogenous perturbation. This model also identifies a critical point in cytosolic tryptophan abundance beyond which descent into the pathological steady-state is inevitable. If, however, means can be discovered to return cytosolic tryptophan below the critical point, return to the normal physiological steady-state is assured. Testing this hypothesis for any cell type requires only labelled tryptophan, a means to measure cytosolic tryptophan and kynurenine, and the standard tools of tracer kinetics.

## 1. Introduction

Diagnostic measurements for a given disease are most definitive, most specific, and most useful when the measurements are closely related to the molecular basis of the disease. Compare, for example, the ancient diagnosis of diabetes based on excessive urine output versus a diagnosis based on the measurement of plasma glucose and plasma insulin. For a disease like myalgic encephalomyelitis/chronic fatigue syndrome (ME/CFS), whose mechanistic basis is unknown, a specific diagnostic is difficult to identify. Medicine thus resorts to complex pattern recognition in lists of symptoms [1,2], statistical principal component analysis [3,4,5,6], advanced physical measurements of yet unproven specificity [7], and even, when just a few thousand critical cells are dysfunctional and standard blood measurements are therefore unremarkable, the unhelpful assertion that patient is “not ill”. Here, we aim to advance the search for underlying mechanisms by proposing a new class of theoretical models for ME/CFS and demonstrating that one specific member of that class, the IDO metabolic trap, can reproduce important features of the disease and promises to be experimentally testable.

The idea of a metabolic trap arises from the well-established concept of bistability in nonlinear systems. A system, such as a metabolic pathway, is said to be nonlinear if a doubling of its input does not yield a doubling of its output. Saturation of enzymatic catalysis, characterized by the classic Michaelis-Menten equation, is one source of biological nonlinearity. A more important source of biological nonlinearity is feedback control whether it be transcriptional regulation, allosteric control mediated by binding of a specific metabolite, post-translational modification mediated by, for example, a kinase, or physiological regulation mediated by hormones, cytokines, and their receptors. Importantly, some nonlinear systems, unlike all linear systems, can settle into multiple different steady-states, depending on external conditions or perturbations. This is called bistability. If one of those steady-states is pathological, we refer to it as a metabolic trap because organisms are vulnerable to external perturbations that precipitate a shift from normal physiology to pathophysiology, a shift that is not easily reversed. If we now turn to the specific case—the IDO metabolic trap developed here—we find that this new theoretical model rests on three ideas: (1) the potential importance of common damaging mutations, (2) a possibly detrimental aspect of the phenomenon in enzyme kinetics known as substrate inhibition, and (3) the bistable metabolic system that can result.

Genetics must hold clues to ME/CFS because, like other chronic diseases, there is evidence that this disease can run in families, but it is clearly not a disease one has at birth. Rather, there appears to be a genetic propensity that lies hidden until a particular collection of triggering circumstances arises in the patient’s microbial, dietary, micronutrient, physiological, emotional or physical environment. One clue that distinguishes ME/CFS from most other chronic diseases is its long history epidemics, outbreaks or clusters [8]. Historically, it has been assumed that the effect of common genetic variations on phenotype is small [9]. Outbreaks or epidemics of a noncontagious disease raise the possibility that genetic predisposition to ME/CFS is very common in the population and that the disease has low penetrance only because the initiating triggers are multifactorial, and those pathogenic *combinations* of triggers are, themselves, rare. Outbreaks are then explained by a geographically localized combination of factors superimposed on a genetic predisposition that is common in the population. Thus, it is the existence of ME/CFS outbreaks that pointed to the potential importance of common damaging mutations. This inference from the existence of outbreaks does *not* limit the model to patients who became sick as part of an outbreak.

A second foundational idea for the IDO metabolic trap model is the phenomenon of substrate inhibition. In contrast to the widely understood Michaelis-Menten kinetics, there are enzymes for which velocity decreases, rather than saturates, at substrate concentrations 3- to 10-fold above the substrate’s K_M_. This unusual behavior has been a part of the enzyme kinetics literature for decades [10,11,12]. One of the giants of enzyme kinetics, W.W. Cleland, took the position that substrate inhibition was almost always a nonphysiological phenomenon [13]. What he meant by “nonphysiological” was that the phenomenon was most often observed when reactions were run opposite the physiological direction with the normal intracellular products as substrates, and thus were examples of product inhibition. Subsequent experience has shown that at least 80 enzymes demonstrate this behavior [10,14]. A recent review [14] describes several desirable consequences of substrate inhibition such as stabilization of product formation in the context of large swings in the substrate (tyrosine hydroxylase), and allosteric regulation (phosphofructokinase). Nevertheless, tryptophan inhibition of the particular enzyme, human indoleamine-2,3-dioxygenase 1 (IDO1), that is central to the model presented here, has been demonstrated in multiple laboratories over multiple decades [15,16,17,18] running in the physiological direction with tryptophan and oxygen as substrates and n-formyl-L-kynurenine as a product. While the phenomenon of IDO1 substrate inhibition is well-established, its mechanism remains a matter of scientific debate [17,18,19]. The model presented here explores the possibility that substrate inhibition of IDO1 has a dark side and may be involved in the pathogenesis of ME/CFS.

A third concept, bistability as a feature of some enzymatic systems, has been studied extensively. Even 40 years ago an itemization of biological oscillators included hundreds of published papers [20]. Later, an oxidase in vitro was shown to exhibit all three of the major dynamic features of nonlinear systems: bistability, limit cycle oscillations, and chaos [21]. And in recent years, bistability in metabolism is enjoying a resurgence of research interest [22,23,24]. While the IDO metabolic trap model is focused on metabolic bistability, other research in ME/CFS has sought bistability at cell biological and physiological levels of organization [25,26]. Some ME/CFS patients experience the onset of the disease as a switch being thrown, and researchers with backgrounds in nonlinear system theory are therefore drawn to theories that involve bistability [27].

This paper is largely theoretical. Its aim is to formulate an internally consistent hypothetical mechanistic model of the etiology of ME/CFS and to propose an experiment capable of rejecting or corroborating this model. The IDO metabolic trap model represents a new way to think about ME/CFS. It is based (1) on the existence of common damaging mutations in human IDO2, (2) on the well-studied kinetic characteristics of IDO2 and IDO1 including substrate inhibition of IDO1, and (3) on the demonstrable bistability that results when these enzymes are expressed in cells that rely on the large neutral amino acid transporter, LAT1, to import tryptophan, which is their carbon-containing substrate. The model has considerable explanatory power because of the cell types expressing these enzymes, and a relatively straightforward experimental test based on tracer kinetics is proposed. If the IDO metabolic trap is found to be a feature of ME/CFS immune cells, a strong basis for the development of a specific ME/CFS diagnostic will have been discovered.

## 2. Materials and Methods

Public databases: Starting from the inference that predisposing damaging mutations must be common in order to account for the existence of ME/CFS outbreaks, a search of public (NCBI dbSNP) [28] and purpose-built ME/CFS (see Acknowledgments) databases for common damaging mutations in genes coding for proteins, particularly enzymes and transporters, involved in energy metabolism was undertaken. Results were displayed in standard genome browsers, IGV [29] and UCSC [30]. Allele frequencies for alternate alleles were obtained from dBSNP [28] and large-scale genome sequencing projects cited therein. Damaging mutations were identified based on standard prediction algorithms, PROVEAN [31], SIFT [32], and PolyPhen-2 [33] as well as published reports [34].

Bioinformatics: To generate a table of genes that are damaged in at least 85% of the severely ill ME/CFS patients, we first filtered all the variants found in the OMF END ME/CFS dataset (see Acknowledgments) considering only mutations that were not excluded by six standard criteria. (1) Indel genotypes from two or more loci conflict in at least one sample. (2) The site contains an overlapping indel call filter. (3) Locus GQX is less than 15 or not present. (4) The fraction of base calls filtered out at a site is greater than 0.4. (5) The sample SNV strand bias value exceeds 10. (6) The locus depth is greater than 3× the mean chromosome depth. Mutations were deemed damaging if the mutation received a score less than or equal to −1.82 by PROVEAN, less than or equal to 0.05 by SIFT, and greater than or equal to 0.95 by PolyPhen-2. Mutations were then grouped by gene, and a table was compiled of (82) genes such that at least 85% of the SIPS patients have one or more damaging mutations in that gene.

A second table containing all enzymes and transferases/transporters (208 total) damaged by common mutations (AF > 0.03) was generated by joining all the filtered non-synonymous mutations considered damaging by the above definition with the data extracted by BRENDA [35], KEGG [36], and TCDB [37].

Mechanistic kinetic modeling: Nonlinear kinetic models were formulated in the ProcessDB software (Integrative Bioinformatics, Inc, Mountain View, CA, USA) [38]. ProcessDB implements the CVODE algorithm [39] for a numerical solution of the differential equations. Steady-state solutions were graphed in the Origin 2019 software (OriginLab Corp., Northampton, MA, USA). Full equations and parameter values are provided in the text and can be implemented in any general-purpose differential equation solver. Parameters and some rate laws were obtained from expert reports on the recombinant human enzymes, IDO1 and IDO2 [17,18,40,41].

## 3. Results

Examination of the two tables of candidate genes described in Bioinformatics Methods revealed common mutations in 208 enzymes and transporters. Of these, eight had more than one common damaging mutation. Reasoning that multiple common damaging mutations would increase the probability of a damaged protein product, we turned our initial attention to this subset.

### 3.1. Common Mutations in IDO2

Given the hypometabolic phenotype of ME/CFS, our search for common damaging mutations began with genes coding for enzymes involved in energy metabolism. Of the eight enzymes with multiple common damaging mutations (allele frequency > 0.03) IDO2 stood out because it has four such mutations and because it is one of the enzymes catalyzing the first step in the kynurenine pathway. Classically, the kynurenine pathway is considered the “de novo” pathway for the synthesis of nicotinamide adenine dinucleotide (NAD^+^), a molecule essential for transferring reducing equivalents from central carbon metabolism to the mitochondrial electron transport chain and thus powering oxidative phosphorylation. Table 1 lists both the common and rare mutations in IDO2 that are considered damaging by the PROVEAN, SIFT, and PolyPhen-2 prediction algorithms.

The two most common damaging mutations, R248W and Y359STOP, are known to abolish enzyme activity in an in vitro cell kynurenine production assay [34]. While the corresponding experiments for I140V, S252T, and N257K have not been reported, the SIFT, PROVEAN, and POLYPHEN predictions are suggestive and could contribute compound heterozygosity for individuals who are merely heterozygous for R248W or Y359STOP. There are no such common mutations in IDO1 or in the remainder of the kynurenine pathway.

Importantly, the IDO metabolic trap hypothesis does *not* propose that these common damaging mutations in IDO2 are causal for ME/CFS. The only requirement for a predisposing mutation is that it is present in ME/CFS patients. On this hypothesis, population allele frequencies recorded in Table 1 should be statistically significantly different from the corresponding allele frequencies in ME/CFS patient populations. Considering the extremely high variant allele frequencies in the general population (Table 1), achieving statistical significance may require targeted sequencing of the IDO2 gene in a very large ME/CFS patient population.

### 3.2. Consequences of Non-Functional IDO2

Since IDO2 and IDO1 catalyze the same reaction at the beginning of the kynurenine pathway, we are obligated to ask how a non-functional IDO2 has any metabolic impact. After all, IDO1 is unimpaired by the IDO2 damaging mutations and is perfectly capable of converting tryptophan to N-formyl kynurenine (NFK). One possibility is that IDO2 can do something that IDO1 cannot, which leads to a more detailed consideration of IDO1 and IDO2 enzyme kinetics.

Figure 1 plots the flux of NFK production as a function of substrate concentration for IDO1 and IDO2. These graphs are calculated using kinetic parameters reported in the literature (see Methods) for the normal Michaelis-Menten behavior of human IDO2 and the substrate inhibited behavior of human IDO1.

Three important points emerge from examination of Figure 1. First, IDO1 is a substrate inhibited enzyme with K_i_(Trp) = ~50 µM and IDO2 is characterized by normal Michaelis-Menten kinetics. Second, IDO1 is a high-affinity enzyme with K_M_(Trp) = ~5 µM and IDO2 is a low-affinity enzyme with K_M_(Trp) = ~100 µM. Third, when both enzymes are functional, the total IDO-mediated conversion of Trp to NFK is monotonically increasing and approximately Michaelis-Menten in form over a wide range of substrate (Trp) concentration (panel a), but when IDO2 is functionally impaired by a common damaging mutation (panel b), total IDO flux decreases when substrate [Trp] increases above ~30 µM. Thus, what IDO2 can do that IDO1 cannot is to catalyze Trp + O_2_
→ NFK even when [Trp] > 200 µM.

These features of IDO kinetics can, when the supply of substrate Trp does not decrease when IDO1 is substrate inhibited, create an untoward metabolic situation that can be referred to as a metabolic trap. To provide a quantitative description of this IDO metabolic trap, we can consider the abbreviated metabolic model illustrated in Figure 2.

### 3.3. A Mechanistic Model of the IDO Metabolic Trap Reveals Bistability

Tryptophan is an essential amino acid; for humans, its only source is dietary. Transport across the intestinal epithelium is dominated by the LAT2 transporter while crossing the capillary endothelium, the blood-brain barrier, and the serotonergic neuronal plasma membrane is mediated by the heterodimeric transporter, LAT1. LAT1 is an obligate antiport; it transports a large neutral amino acid (often leucine) out of the cell each time it transports tryptophan in. An important feature of LAT1-mediated transport is that K_s_ values for amino acid uptake are in the 15 µM range, while K_s_ values for amino acid export are in the mM range [42,43].

To capture this kinetic asymmetry, a reversible Michaelis-Menten rate law [44] for the flux (molecules/min/cell) can be written as:(1)$$J_{LAT1}=\frac{V_{mf}\left(1-\frac{T_{cyto}}{T_{ECF}K_{eq}}\right)\left(\frac{T_{ECF}}{K_{s}^{T_{ECF}}}\right)}{1+\frac{T_{ECF}}{K_{s}^{T_{ECF}}}+\frac{T_{cyto}}{K_{s}^{T_{cyto}}}+\frac{A}{K_{i}^{A}}}$$
where $J_{LAT1}$ is the net flux of tryptophan (positive means the net flux is into the neuron cytosol from the extracellular space), $V_{mf}$ is the maximal forward (into the neuron) velocity of LAT1 transport, $T_{cyto}$ is the tryptophan abundance (molecules/cell) in the neuron cytosol, $T_{ECF}$ is tryptophan abundance in the extracellular fluid, $K_{eq}$ is the equilibrium constant for LAT1 transport, $K_{s}^{T_{ECF}}$ is the substrate constant for tryptophan uptake, $K_{s}^{T_{cyto}}$ is the substrate constant for tryptophan export, A is the abundance of other large neutral amino acids (either in ECF or cytosol) competing with tryptophan for transport via LAT1, and $K_{i}^{A}$ is the inhibition constant for those competing amino acids.

Next, we need a quantitative expression for the flux through the substrate-inhibited enzyme, IDO1. This can be written using a standard substrate inhibition rate law:(2)$$J_{IDO1}=\frac{k_{cat}^{IDO1}IDO1_{cyto}T_{cyto}}{K_{M}^{Trp}+T_{cyto}\left(1+\frac{T_{cyto}}{K_{i}^{Trp}}\right)}$$
where $J_{IDO1}$ is the flux (molecules/min/cell) of Trp conversion to NFK catalyzed by IDO1, $k_{cat}^{IDO1}$ is the catalytic constant for IDO1, $IDO1_{cyto}$ is the abundance (molecules/cell) of cytosolic IDO1, $T_{cyto}$ is the abundance of cytosolic tryptophan, $K_{M}^{Trp}$ is the IDO1 Michaelis constant for tryptophan, and $K_{i}^{Trp}$ is the substrate-inhibition constant for Trp inhibition of IDO1.

A full kinetic model would also include rate laws for IDO2, for tryptophan hydroxylase (TPH1 or TPH2 depending on the cell type being modeled) and for protein translation and proteolysis. Here, we can consider the case where IDO2 is disabled by common damaging mutations, where serotonin production is negligible compared to the kynurenine pathway flux, and where (in a steady-state) the flux of protein synthesis is equal to the proteolytic flux and thus there is no net flux between cytosolic tryptophan and cellular protein. These must be relaxed when it comes time for data analysis, but to establish bistability only the principal input of Trp, $J_{LAT1},$ and the principal output, $J_{IDO1}$, need be considered.

Thus, we can write the differential equation for cytosolic tryptophan as:(3)$$\frac{dT_{cyto}}{dt}=J_{LAT1}-J_{IDO1}$$
where $J_{LAT1}$ and $J_{IDO1}$ are as defined above. This is the simplest possible mathematical model of the IDO metabolic trap. To obtain the steady-states of this model requires only that we set $\frac{dT_{cyto}}{dt}=0,$ and solve for $T_{cyto}$. This would require solving a cubic algebraic equation, which is entirely feasible, but a more straightforward approach is to solve for $T_{cyto}$ graphically as illustrated in Figure 3.

In Figure 3 the horizontal axis represents the cytosolic Trp abundance on a logarithmic scale. This is why the LAT1 flux decreases as Trp increases. Indeed, at the largest values of $T_{cyto}$, the LAT1 flux becomes negative indicating that the net LAT1 flux is directed out of the cell. The increasing and then decreasing shape of the IDO1 flux curve will be familiar from Figure 1 and is caused by the substrate inhibition of IDO1 by tryptophan. Since setting $\frac{dT_{cyto}}{dt}=0$ requires $J_{LAT1}=J_{IDO1}$, the three steady-states correspond to the intersections (labeled A, B, and C) of the blue and red curves in Figure 3.

Steady-states labeled A and C are stable, and the steady-state labeled B is unstable. These conclusions can be drawn directly from Figure 3. Consider the steady-state labeled A. If stochastic variation increases cytosolic Trp abundance then IDO1 outflux will be greater than LAT1 influx, and $T_{cyto}$ will decrease until it returns to point A. If, instead, the initial variation decreases $T_{cyto}$, LAT1 influx will become greater than IDO1 outflux, and $T_{cyto}$ will increase until it again returns to point A. The same is true for the steady-state labeled C.

But steady-state B is different. It is unstable because stochastic variation in $T_{cyto}$ results in changes to influx and outflux that drive the system to either steady-state A or steady-state B. For this reason, point B defines what is called a *critical point*. If cytosolic tryptophan exceeds the value defined by this critical point, the system falls inexorably into steady-state C. This extremely simple model is thus *bistable*. It is capable of both a normal physiological steady-state (A) in which kynurenine production is supported by LAT1-mediated import of tryptophan and cytosolic tryptophan is ~2 × 10^7^ molecules/cell, and a pathological steady-state (C) in which kynurenine production is nearly abolished and cytosolic tryptophan is ~5 × 10^9^ molecules/cell, more than two orders of magnitude greater than normal for a cell expressing the kynurenine pathway.

### 3.4. An Experimental Design to Test the Trap Hypothesis

Every hypothesis needs an experimental test capable of rejecting or corroborating it. For the IDO metabolic trap hypothesis, the natural test is based on metabolic tracer kinetics [45] and the well-developed set of computational tools [46,47,48,49] for analysis of the resulting tracer data. Here, the natural tracer is tryptophan labeled with ^14^C or ^13^C in its indole ring and, optionally, in its alpha and carboxyl carbons as well. Tryptophan labeled with either isotope can be used in animal studies and freshly isolated or cultured cells, while the stable isotope, ^13^C, is, today, widely preferred for human metabolic studies. For isolated cells, the choice usually depends on local expertise and training.

A useful test requires methods for isolating intracellular compounds after exposing the cells to ^13^C-tryptophan for a specified loading period. At specified times thereafter multiple cold washes both stop the cellular reactions and remove the extracellular medium so that intracellular measurements are not contaminated with extracellular unlabeled or labeled tryptophan or secreted serotonin or kynurenine. It would then be possible to measure cytosolic ^13^C-tryptophan as well as the incorporation of labeled tryptophan into cellular kynurenine and serotonin. If transients in ^13^C-kynurenine and ^13^C-serotonin are measured after removing the tryptophan tracer from the medium. A great advantage of stable isotope kinetics is that the corresponding endogenous ^12^C compounds are measured simultaneously and quantified separately by the liquid chromatography-mass spectrometry workflow and thus further constrain the computational analysis of the data. The hypothesis will be corroborated if the flux through IDO to kynurenine is substantially reduced in cells from ME/CFS patients compared to healthy control subjects. The hypothesis will be rejected for the cell type tested if there is no difference in the IDO fluxes measured in the two experimental groups.

## 4. Discussion

The IDO metabolic trap hypothesis for ME/CFS thus suggests that four cell types are at risk of being driven into the pathological steady-state C (Figure 3): (1) antigen-presenting cells (such as dendritic cells and macrophages), (2) serotonergic neurons in the midbrain raphe nuclei, (3) serotonin-producing enterochromaffin cells in the intestinal mucosa, and (4) melatonin-producing pinealocytes. This risk, according to the hypothesis, is magnified by the absence or dysfunction of the backup enzyme, IDO2. A cell in the pathological steady-state C can be described as being in the IDO metabolic trap.

### 4.1. Consequences of the IDO Metabolic Trap

Consequences of being in this abnormal steady-state depend on many factors. For example, if the normal flux through the kynurenine pathway is the major route of tryptophan oxidation/removal, then cytosolic tryptophan will increase dramatically if IDO1 becomes substrate inhibited. In turn, increased cytosolic tryptophan will drive excessive serotonin production in cells, like serotonergic raphe neurons, that express TPH2 (human chromosome 12) [50] with its normal Michaelis-Menten kinetics [51], or may result in decreased serotonin and melatonin production in cells, such as antigen presenting cells, enterochromaffin cells, and pinealocytes that express the classical “peripheral” tryptophan hydroxylase, TPH1 (human chromosome 11) [50]. This is because TPH1 is, itself, substrate inhibited at high concentrations of its substrate, tryptophan [51,52].

Even if serotonin and melatonin production are unperturbed, the absence of kynurenine and its metabolites can have untoward effects. For example, kynurenine is spontaneously converted to trace condensation products called TEACOPs that are potent activators of the aryl hydrocarbon receptor (AHR), a transcription factor that controls the development of T_reg_ cells [53]. In addition, kynurenine is deaminated by kynurenine aminotransferase and the resulting kynurenate is a potent neuroprotective alpha 7-nicotinic acetylcholine receptor antagonist [54].

If instead we follow the main branch of the kynurenine pathway, additional consequences of insufficient kynurenine production appear. First, the spontaneous synthesis of quinolinate from 2-amino-3-carboxymuconate semialdehyde will decrease. Not only is quinolinate itself neuroexcitatory, but it is also the precursor of nicotinate mononucleotide, which is an important source of NAD^+^, especially if dietary nicotinate is limited [55]. Second, the enzymatically catalyzed route from 2-amino-3-carboxymuconate semialdehyde leads to picolinate, which is widely described as a neuroprotectant. Perhaps importantly, picolinate production is potently inhibited by glycolytic triose phosphates [56]. Thus, if cytosolic NAD^+^ is insufficient to maintain a high glycolytic flux through glyceraldehyde 3-phosphate dehydrogenase, the resulting increase in triose phosphates will divert the kynurenine pathway away from neuroprotective picolinate and toward quinolinate/NAD^+^. This regulatory cross-talk between glycolysis and the kynurenine pathway will certainly fail in trapped cells.

### 4.2. IDO1-LAT1 Critical Point as the Threshold of Chronic Disease

In Figure 3, the unstable steady-state, B, is the critical point in cytosolic tryptophan concentration. Mathematically, it is the threshold above which the solution of the differential equation switches to the alternate attractor. Practically, the critical point marks the cytosolic tryptophan concentration above which the cell is destined for the pathological low-kynurenine steady-state.

It may be important to recognize that at some increased level of LAT1 expression, only one steady-state is possible. And that steady-state is the pathological one. Conversely, there are decreased levels of LAT1 expression that are only compatible with a normal physiological steady-state.

Another implication of the critical point is that therapeutic perturbations need only reduce cytosolic tryptophan below the critical point. Below that point a return to the normal physiological steady-state is inevitable. It must be kept in mind, however, that the trap persists, and subsequent triggers can cause relapse.

It remains to be determined whether this bistable system is also capable of limit cycle oscillations or deterministic chaos, which could explain relapsing-remitting forms of ME/CFS and the seeming impossibility of multiple metabalomics studies converging on the same plasma signature.

Again, we emphasize that all these points are based on a hypothesis. Without experimental corroboration of that hypothesis and, optimally, a clinical trial, these statements about therapy are not actionable. They are intended to provoke research.

### 4.3. Underlying Assumptions

Deficits in kynurenine production and decreases in the abundance of its downstream metabolites (kynurenate, anthranilate, picolinate, quinolinate, NAD^+^, and others) will occur in any cell whose IDO1 becomes substrate inhibited. Implicit in this scenario is that IDO2 activity provides insufficient backup at high cytosolic tryptophan. This, of course, is the rationale for the attention paid to common damaging mutations in IDO2. Similarly, the IDO metabolic trap cannot occur in any cell type that also expresses tryptophan-2,3-dioxygenase (TDO).

Hypo- or hyper- synthesis of tryptophan-derived serotonin or melatonin depends on the extent to which cytosolic tryptophan abundance is increased relative to the K_M_ and K_i_ values of tryptophan hydroxylase. This, in turn, depends on the assumption that the normal kynurenine pathway flux is much greater than the normal tryptophan hydroxylase flux. This assumption can be tested in any given cell type with the tracer kinetic experiment proposed in Section 3.4.

A final assumption of the IDO metabolic trap model is that tryptophan uptake by LAT1 has the features described in the literature [42], namely its kinetic asymmetry. The K_M_ for tryptophan on the outside aspect of the transporter is reportedly in the uM range while the K_M_ on the cytosolic aspect is 1000-fold greater. Any other allosteric control of the transporter that maintains tryptophan transport despite decreased utilization by the kynurenine pathway would also satisfy this assumption. This feature of the model deserves further attention because all cytosolic large neutral amino acids (including kynurenine) can play both activator and inhibitor roles for LAT1-mediated tryptophan import.

### 4.4. Potential Role of Bistability and Substrate Inhibition in Chronic Disease

While it is possible that the IDO metabolic trap lays bare the etiology of ME/CFS, the probability that this is so is small. When we search for common damaging mutations in the human genome, we, by definition, find many. Consequently, the hypothesized increase in damaging alleles in ME/CFS will be difficult to demonstrate without a much larger population of carefully diagnosed ME/CFS patients whose genome sequences are available. For a given trap hypothesis, such as the IDO metabolic trap, it is likely that targeted sequencing of the relevant gene is a more efficient and cost-effective approach. Furthermore, one could question whether a classical genetic family study would be fruitful in this context because the presence of a predisposing mutation need not correlate with the presence of disease. If a common damaging mutation is predisposing (not causal), its allele frequency in a given population may be far greater than the prevalence of the disease for which it is predisposing. The mutation(s) will have low penetrance because it is one or more rare combinations of environmental circumstances that trigger the disease and determines its prevalence. If this is true, the fraction of the world’s population at risk for ME/CFS may be vastly greater than its 2% estimated prevalence of the disease.

Another model of chronic disease is the comprehensive healing cycle (HC) model of Naviaux [57]. The HC model emphasizes that knowing the causes of ME/CFS (or other chronic diseases) will not point to a cure. Instead, ME/CFS is seen as a failure to pass the third and final checkpoint in the healing cycle [57]. On this theory, diseases like ME/CFS are characterized by abnormal cell-cell communication that, in turn, is caused by dysfunctional G-protein coupled receptors (especially purinergic receptors) and their regulatory metabokine networks. The result is the failure of the healing cycle and, thus, chronic disease. It is in this sense that the HC model views chronic disease as “a systems problem that maintains disease”.

The IDO metabolic trap is a prototype for a different class of systems-level problems. Like the HC model, the trap model implies that chronic disease is not the result of chronic exposure to a pathogen or toxin. Unlike the HC model, the trap model does not suggest a failure of healing or, indeed, a failure of anything at all. Instead, the trap hypothesis identifies substrate inhibition, an inherent and well-studied feature of some enzymes that can, in response to pathogenic triggers, unmask metabolic bistability. Bistability is a phenomenon unique to nonlinear dynamics that has been studied for decades [23,24,58,59]. The essential fact of bistability is that it is possible for a bistable system to be driven into an alternative (e.g., pathological) steady-state and be maintained in that chronic disease state long after the trigger is removed and *without requiring a chronic infection or chronic exposure to a toxin*. Furthermore, nothing need be broken or dysfunctional. The possibility of this chronic disease state is inherent in the detailed molecular mechanism of substrate inhibition.

Because substrate inhibition is relatively common [10], the features of the IDO metabolic trap may appear in other pathways and constitute the mechanisms of other chronic diseases. As an example, tyrosine hydroxylase, the first and rate-determining step in dopamine synthesis, is also substrate inhibited and could provide an alternative explanation for low dopamine production in Parkinson disease or for insufficient production of catecholamines in other cell types.

To summarize, the IDO metabolic trap hypothesis for ME/CFS suggests that substrate inhibition of IDO1 creates the possibility of metabolic bistability in cells expressing the kynurenine pathway. Transition from the normal physiological steady-state to the alternative steady-state can be initiated by any trigger that increases cytosolic tryptophan concentration above the critical point (point B in Figure 3). The alternative, or pathological, steady-state is characterized by insufficient kynurenine production from tryptophan, and consequent impairments in the central nervous system, gastrointestinal, and immune function, as well as energy metabolism. Depending on the relative expression and kinetics of other tryptophan-dependent enzymes in enterochromaffin cells and pinealocytes, production of other tryptophan metabolites (e.g., serotonin and melatonin) may be pathogenically altered as well. For any given cell type, this hypothesis is testable using labelled tryptophan combined with standard tracer kinetic analysis [48].

## Figures and Tables

**Figure 1 diagnostics-09-00082-f001:**
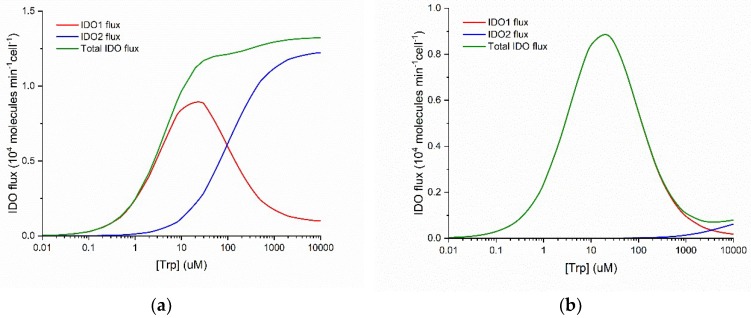
Differences in IDO1 (red) and IDO2 (blue) enzyme kinetics as functions of Trp concentration. Total IDO flux (green) is the sum of the IDO1 and IDO2 fluxes. (**a**) Wild type situation with IDO1 and IDO2 having comparable V_max_ values; (**b**) Fluxes when IDO2 flux is 90% reduced, for example, by the homozygous common damaging mutation, R248W.

**Figure 2 diagnostics-09-00082-f002:**
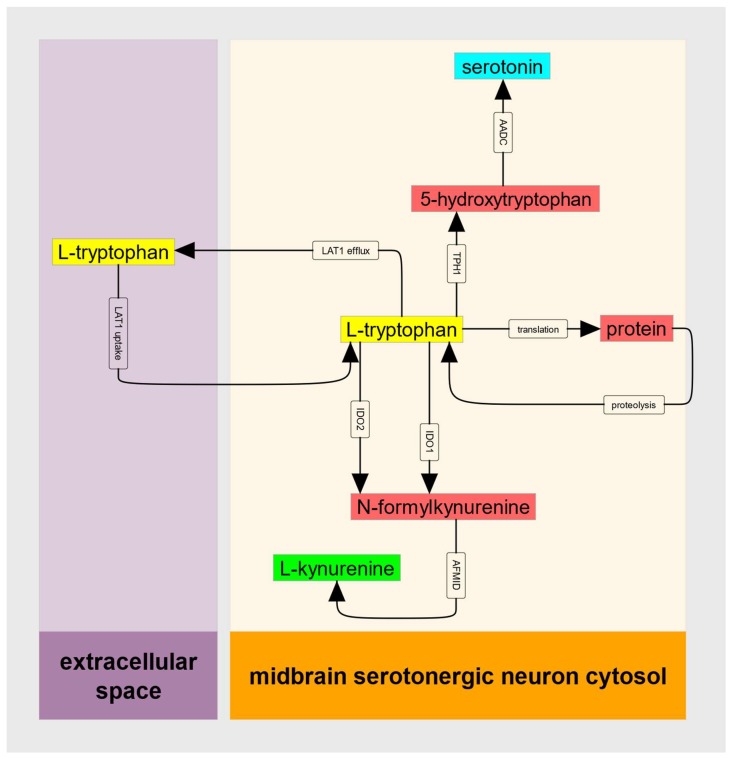
Diagram of the kinetic model of the IDO metabolic trap. Colored rectangles represent molecules in either extracellular space or serotonergic neuron cytosol. Arrows represent processes including transport and biochemical reactions. LAT1 = large neutral amino acid transporter (SLC7A5:SLC3A2), IDO = indoleamine-2,3-dioxygenase, AFMID = arylforamidase, TPH = tryptophan hydroxylase, AADC = aromatic amino acid decarboxylase.

**Figure 3 diagnostics-09-00082-f003:**
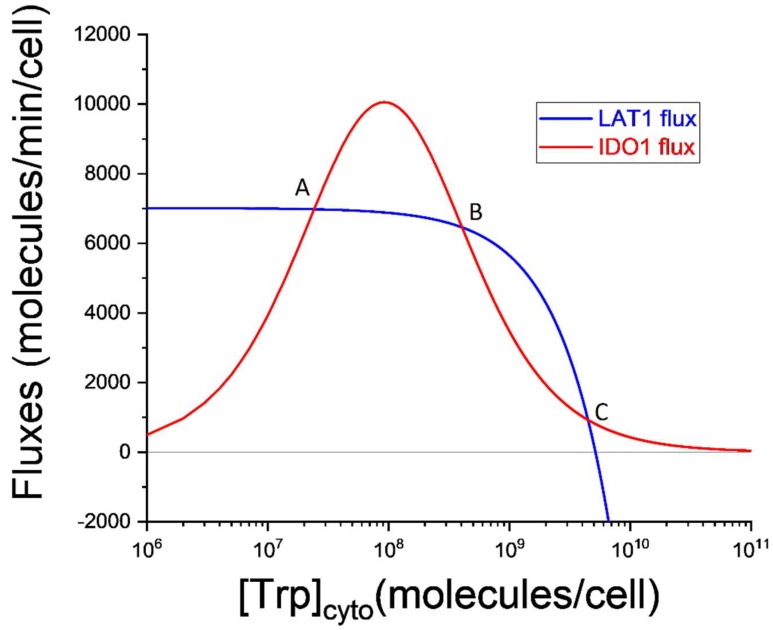
Multiple steady-states in the simplest model of LAT1 tryptophan (Trp) transport and IDO1-mediated Trp oxidation. Horizontal axis: cytosolic Trp abundance. Vertical axis: Fluxes (molecules·min^−1^·cell^−1^) cellular Trp influx (blue) carried by the LAT1 membrane transporter, and Trp removal (red) catalyzed by IDO1. Three possible steady-states are defined by the three points (A, B, and C) where the two fluxes are equal. Numerical parameter values are: $k_{cat}^{IDO1}$ = 84 molecules·min^−1^ IDO1·molecule^−1^, $IDO1_{cyto}$ = 208 IDO1 molecules/cell, $K_{M}^{Trp}$ = 3.4 × 10^7^ molecules/cell, $K_{i}^{Trp}$ = 2.5 × 10^8^ molecules/cell, $V_{mf}$ = 1.2 × 10^8^ molecules·min^−1^·cell^−1^, $T_{ECF}$ = 1.5 × 10^9^ molecules/cell, $K_{eq}$ = 3.43, $K_{s}^{T_{ECF}}$ = 2.2 × 10^10^ molecules/cell, $K_{s}^{T_{cyto}}$ = 2 × 10^9^ molecules/cell, A = 5 × 10^6^ molecules/cell, and $K_{i}^{A}$ = 4.3 × 10^3^ molecules/cell.

**Table 1 diagnostics-09-00082-t001:** Common and rare mutations in IDO2 identified as damaging ^3^.

Row Label	R248W	Y359STOP	I140V	S252T	N257K
dbSNP ID	rs10109853	rs4503083	rs4736794	rs35212142	rs774492001
Allele ref > alt	C > T	T > A	A > G	T > A	C > G
exon	9	11	5	9	10
Min pop AF ^1^	0.418	0.220	0.0746	0.0100	0.000017
Max pop AF ^2^	0.487	0.230	0.160	0.0390	0.000020
SIFT	damaging	nonsense	damaging	damaging	damaging
PROVEAN	deleterious	nonsense	neutral	deleterious	deleterious
POLYPHEN	probably damaging	nonsense	possibly damaging	probably damaging	probably damaging

^1^ minimum alternate allele frequency (expressed as a fraction) reported [28] for any sampled population, ^2^ maximum alternate allele frequency reported for any sampled population, ^3^ ‘damaging” means the enzyme encoded by the mutant protein is either known or predicted to be catalytically impaired.

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
