# Peer review of "The IDO Metabolic Trap Hypothesis for the Etiology of ME/CFS"

_diagnostics, 2019, doi:10.3390/diagnostics9030082_

Round 1

Reviewer 1 Report

Kashi et al. present a fascinating hypothesis, supported by a literature review and mathematical model, to propose a role for "metabolomic traps" involving IDO and tryptophan pathways, as relevant to ME/CFS pathology. The authors generously provide guidance to experimentalists on how to test their hypothesis in the laboratory. Given the compelling metabolomic results emerging from laboratories internationally, this is a timely contribution, to give pause and reflect on the implications of the lab-based insights on ME/CFS.

Prior to publication, can the authors please consider -

(1) Introduction: Between the 1st and 2nd paragraphs, can you please add a short paragraph explaining the broad "metabolic trap" concept.

(2) "Table 1. Common and rare mutations in IDO2 identified as damaging" - generally "damaging", or in the ME/CFS context? - please clarify.

(3) From line 281 - "A great advantage of stable isotope kinetics is that the corresponding endogenous 12C compounds are measured simultaneously by the liquid chromatography-mass spectrometry workflow and thus constrain the computational analysis of the data" - Clarification please - the way I read this is that there would be less constraint for the analyses?

(4) Optional - As the content is heavy in high-powered biochemical and mathematical concepts, can the authors please consider adding a glossary/definitions Box (Box 1) to the manuscript. Within this, a reminder on the fundamentals of Michaelis-Menten kinetics, will be very helpful (a wide readership can be expected for this Special Issue of Diagnostics).

Author Response

Kashi et al. present a fascinating hypothesis, supported by a literature review and mathematical model, to propose a role for "metabolomic traps" involving IDO and tryptophan pathways, as relevant to ME/CFS pathology. The authors generously provide guidance to experimentalists on how to test their hypothesis in the laboratory. Given the compelling metabolomic results emerging from laboratories internationally, this is a timely contribution, to give pause and reflect on the implications of the lab-based insights on ME/CFS.

Thank you for this highly positive assessment. Having, ourselves, tried to follow our “generously provide[d]” experimental guidance, we can say it’s not that easy. But that’s another manuscript.

Prior to publication, can the authors please consider -

(1) Introduction: Between the 1st and 2nd paragraphs, can you please add a short paragraph explaining the broad "metabolic trap" concept.

This was a good idea. As requested, additional explanation of the general metabolic trap concept has been added to paragraph 2.

(2) "Table 1. Common and rare mutations in IDO2 identified as damaging" - generally "damaging", or in the ME/CFS context? - please clarify.

In this context the word “damaging” means that the protein encoded by the mutant gene displays reduced or absent catalytic function. This clarification has been added to the Table legend.

(3) From line 281 - "A great advantage of stable isotope kinetics is that the corresponding endogenous 12C compounds are measured simultaneously by the liquid chromatography-mass spectrometry workflow and thus constrain the computational analysis of the data" - Clarification please - the way I read this is that there would be less constraint for the analyses?

The reviewer’s point is well taken. The sentence has been revised to make clear that the 13C and 12C compounds, while measured simultaneously, are quantified separately and therefore provide additional constraints to the modeling analysis.

(4) Optional - As the content is heavy in high-powered biochemical and mathematical concepts, can the authors please consider adding a glossary/definitions Box (Box 1) to the manuscript. Within this, a reminder on the fundamentals of Michaelis-Menten kinetics, will be very helpful (a wide readership can be expected for this Special Issue of Diagnostics).

We agree that a glossary would be helpful, but since the Diagnostics manuscript template does not offer the possibility of a manuscript Box, we have elected not to delay the publication process further.

Reviewer 2 Report

The hypothesis is a simple one directed at IDO1 and 2. These enzymes are very dependant upon the levels of amino acids which are transported by LAT1. LAT1 is composed of two components a heavy and a light chained protein. The heavy chained protein is competed for by several other light chained proteins, which may alter cellular uptake based upon their competing protein concentrations,Whilst LAT1 is mentioned in the paper some reference to the mechanism may be required.. IDO1 and IDO2 have different cofactors and these included FAD, heme and ttetrahydrobiopterin. There are two different metals involved Copper for one of them and iron for the other. I would seem logical that the mathematics should take some of these interactions into account if these test is to be used on MECFS patients..

Author Response

The hypothesis is a simple one directed at IDO1 and 2. These enzymes are very dependent upon the levels of amino acids which are transported by LAT1. LAT1 is composed of two components a heavy and a light chained protein. The heavy chained protein is competed for by several other light chained proteins, which may alter cellular uptake based upon their competing protein concentrations, Whilst LAT1 is mentioned in the paper some reference to the mechanism may be required. IDO1 and IDO2 have different cofactors and these included FAD, heme and tetrahydrobiopterin. There are two different metals involved Copper for one of them and iron for the other. I would seem logical that the mathematics should take some of these interactions into account if this test is to be used on MECFS patients.

We appreciate the reviewer’s idea that our hypothesis is a simple one. Many in scientific audiences have found it high powered and complex. With luck, the reviewer’s response tells us we have written clearly.

The reviewer’s points about LAT1 are insightful. One of the key features of our model is the reversible Michaelis-Menten formulation of the LAT1 transport rate law. We chose this rate law because it accurately encodes the widely reported (see references 42 and 43) kinetic asymmetry of LAT1 transport – that is, the Km for uptake is two orders of magnitude smaller than the Km for efflux. It seems likely that this feature of LAT1:CD98 supports the increased cytosolic [tryptophan] predicted by the IDO metabolic trap model.

We also appreciate the reviewer’s points concerning the multiple LAT1 heterodimers that can be expressed in a given cell and the resulting differences in competition for transport by amino acids other than tryptophan. We consider that these are adequately modeled as the lumped parameters Vmf, Keq, Ks, and Ki in the equation for LAT1 transport flux on line 307 of the manuscript. Clearly, there are multiple levels of molecular detail at which a transport or enzyme reaction can be modeled, but for our present purposes, the reversible Michaelis-Menten formulation appears adequate.

Finally, additional mechanistic detail, including cofactors, required metals, and even the second substrate, oxygen, will surely be required to investigate regulation of IDO1 and IDO2, and particularly the environmental perturbations that may spring the IDO metabolic trap. We’ve always been interested in the differences among scientists as to what is considered a mechanistic model, and if the trap theory is corroborated experimentally our future models will, as the reviewer states, have to include all these important possibilities as possible means for escape from the trap.